# Systematic review of diagnostic and prognostic models of chronic kidney disease in low-income and middle-income countries

Diego J Aparcana-Granda [1,2] Edson J Ascencio [1,3,4]
Rodrigo M Carrillo Larco [2,5]

For numbered affiliations see end of article.

**Correspondence to**
Dr Rodrigo M Carrillo Larco; rcarrill@ic.ac.uk

## ABSTRACT

**Objective** To summarise available chronic kidney disease (CKD) diagnostic and prognostic models in low-income and middle-income countries (LMICs).

**Method** Systematic review (Preferred Reporting Items for Systematic Reviews and Meta-Analyses guidelines). We searched Medline, EMBASE, Global Health (these three through OVID), Scopus and Web of Science from inception to 9 April 2021, 17 April 2021 and 18 April 2021, respectively. We first screened titles and abstracts, and then studied in detail the selected reports; both phases were conducted by two reviewers independently. We followed the CHecklist for critical Appraisal and data extraction for systematic Reviews of prediction Modelling Studies recommendations and used the Prediction model Risk Of Bias ASsessment Tool for risk of bias assessment.

**Results** The search retrieved 14 845 results, 11 reports were studied in detail and 9 (n=61 134) were included in the qualitative analysis. The proportion of women in the study population varied between 24.5% and 76.6%, and the mean age ranged between 41.8 and 57.7 years. Prevalence of undiagnosed CKD ranged between 1.1% and 29.7%. Age, diabetes mellitus and sex were the most common predictors in the diagnostic and prognostic models. Outcome definition varied greatly, mostly consisting of urinary albumin-to-creatinine ratio and estimated glomerular filtration rate. The highest performance metric was the negative predictive value. All studies exhibited high risk of bias, and some had methodological limitations.

**Conclusion** There is no strong evidence to support the use of a CKD diagnostic or prognostic model throughout LMIC. The development, validation and implementation of risk scores must be a research and public health priority in LMIC to enhance CKD screening to improve timely diagnosis.

## INTRODUCTION

Chronic kidney disease (CKD) is a condition with a large burden globally. Between 1990 and 2017, the health metrics of CKD showed a bleak profile: mortality, incidence and kidney transplantation rates increased by 3%, 29% and 34%, respectively.[1] CKD led to 1.2 million deaths in 2017 and in the best-case scenario,

### Strengths and limitations of this study

► An extensive search was conducted, involving five major databases (Medline, Embase, Global Health, Scopus and Web of Science).

► A comprehensive list of available chronic kidney disease diagnostic and prognostic models and their limitations is provided, which were not previously accounted for in the low-income and middle-income country population.

► This study adhered to Preferred Reporting Items for Systematic Reviews and Meta-Analyses, CHecklist for critical Appraisal and data extraction for systematic Reviews of prediction Modelling Studies and Prediction model Risk Of Bias ASsessment Tool guidelines.

► Meta-analysis was not possible due to the heterogeneity in the measurement of outcomes.

► Additional data sources such as grey literature were not retrieved.

CKD mortality will increase to 2.2 million deaths and become the fifth cause of years of life lost by 2040.[2] CKD reveals disparities between low-income and middle-income countries (LMICs) and high-income countries (HICs). In the period 1990–2016, the age-standardised disability-adjusted life-years due to CKD was the highest in LMIC,[3] where they need to optimise CKD early diagnosis.

Risk scores are a cost-effective alternative for CKD screening and early diagnosis.[4] These equations require less resources and contribute to decision making,[5] and allow screening of large populations.[4] Many of the available CKD risk scores have been developed in HIC,[6–8] and they may not be used in LMIC without recalibration to secure accurate predictions. How many CKD risk scores there are for LMIC, and what their strengths and limitations are, remains largely unknown.[9 10] This limits our knowledge of what tools there are to enhance CKD screening in LMIC.

BMJ

Similarly, this lack of evidence prevents planning research to overcome the limitations of available models. To fill these gaps and to inform CKD screening strategies in LMIC, we summarised available CKD diagnostic and prognostic models in LMIC.

## METHODS

### Protocol and registration

This systematic review and critical appraisal of the scientific literature was conducted following the Preferred Reporting Items for Systematic Reviews and Meta-Analyses guidelines statement[11] (online supplemental table S1). Protocol is available elsewhere[12] and in online supplemental text S1. We followed the CHecklist for critical Appraisal and data extraction for systematic Reviews of prediction Modelling Studies (CHARMS) guidelines.[13 14]

### Information sources

We searched Medline, EMBASE, Global Health (these three through OVID), Scopus and Web of Science from inception to 9 April 2021, 17 April 2021 and 18 April 2021, respectively. The search strategy is available in online supplemental table S2. We also screened the references of relevant systemic reviews[10] and of the selected studies.

### Eligibility criteria

We sought models which assessed the current CKD status (ie, diagnostic) or future CKD risk (ie, prognostic), aiming to inform physicians, researchers and the general population (table 1). Reports could include model derivation, external validation or both. The target population was adults (≥18 years) in LMIC according to The World Bank.[15]

### Study selection

Reports were selected if the study population included people who were from and currently living in LMIC. Cross-sectional (diagnostic models) and longitudinal studies (prognostic models) with a random sample of the general population were included. The outcome was CKD based on a laboratory or imaging test (isolated or in combination with self-reported diagnosis): urine albumin-creatinine ratio, urine protein-creatinine ratio, albumin excretion ratio, urine sediment, kidney images, kidney biopsy or the estimated glomerular filtration rate (eGFR).[12]

Reports had to present the development and/or validation of a multivariable model. On the other hand, reports with LMIC populations outside LMIC, or those including foreigners living in LMIC, were excluded. Reports that only studied people with underlying conditions (eg, patients with diabetes), people with a specific risk factor (eg, alcohol consumption) or a hospital-based population, were excluded. We also excluded models that were developed using machine learning techniques due to their usually poor report of performance metrics, as noted from previous reviews.[16 17] To overcome this limitation, CHARMS and Prediction model Risk Of Bias ASsessment Tool (PROBAST) tools are currently being adapted to machine learning methodology but are yet to be published.[18]

### Data collation

We used EndNote20 and Rayyan[19] to remove duplicates from the search results. We used Rayyan[19] to screen titles and abstracts by two reviewers independently (DJA-G and EJA); discrepancies were solved by consensus. Two reviewers independently (DJA-G and EJA) studied the full length of the reports selected in the screening phase;

| Table 1 | CHARMS criteria to define research question and strategy |
| --- | --- |
| **Concept** | **Criteria** |
| Prognostic or diagnostic? | Both—this review focused on diagnostic and prognostic risk scores for CKD |
| Scope | Diagnostic/prognostic models to inform physicians, researchers and the general population whether they are likely to have CKD (ie, diagnostic) or will be likely to have CKD (ie, prognostic) |
| Type of prediction modelling studies | ▶ Diagnostic/prognostic models with external validation<br>▶ Diagnostic/prognostic models without external validation<br>▶ Diagnostic/prognostic models validation |
| Target population to whom the prediction model applies | General adult population in LMIC. No age or gender restrictions |
| Outcome to be predicted | CKD (diagnostic or prognostic) |
| Time span of prediction | Any, prognostic models will not be included/excluded based on the prediction time span |
| Intended moment of using the model | Diagnostic/prognostic models to be used in asymptomatic adults of LMIC to ascertain current CKD status or future risk of developing CKD. These models could be used for screening, treatment allocation in primary prevention, or research purposes |

Based on the CHARMS checklist.[14]
CHARMS, CHecklist for critical Appraisal and data extraction for systematic Reviews of prediction Modelling Studies; CKD, chronic kidney disease; LMIC, low-income and middle-income country.

discrepancies were solved by consensus. If consensus was not reached, a third party was consulted (RMCL). A data extraction form based on the CHARMS guidelines[14] was developed and not modified during data collation. Data were extracted as presented in the original reports by two reviewers independently (DJA-G and EJA); discrepancies were solved by consensus.

### Risk of bias of individual studies

We used the PROBAST to assess the risk of bias of diagnostic and prognostic models.[20 21] Two reviewers (EJA and DJA-G) independently ascertained the risk of bias of individual reports; discrepancies were solved by consensus or a third party (RMCL).

### Synthesis of results

A qualitative synthesis was conducted whereby the characteristics of the selected models was comprehensively described.[12] Quantitative analysis (meta-analysis) was not conducted because the selected models used different predictors and they had different outcome definitions.

### Patient and public involvement

No patient involved.

## RESULTS
### Reports selection

The search yielded 14 845 reports. After removing duplicates (1462 articles), we screened 13 383 titles and abstracts. Then, 11 reports were selected, 1 of them was not available as full text,[22] and the rest (10 articles) were studied in detail. We excluded one report because the study population was not randomly selected,[23] and another report because it was conducted in an HIC.[24] Additionally, one report was identified by reference searching.[25]

Finally, nine reports (n=61 134) were included in the qualitative synthesis (figure 1).

### General characteristics of the selected reports

Original reports were from Iran,[26] India,[27] Peru,[28] South Africa,[25] two from China[29 30] and three from Thailand[31–33] (online supplemental figure S1). All studies were developed on community-based populations with random sampling (online supplemental table S3).

Overall, Wu *et al* studied the largest sample size (n=14 374) which was a population of workers who underwent health checks[30]; conversely, the smallest sample was studied by Mogueo *et al* (n=902).[25] The oldest data were collected in 1999[26] whereas the most recent study was published in 2018.[26]

The sample size analysed to derive the diagnostic models ranged from 2368[28] to 14 374 people,[30] and from 902[25] to 4940[27] for the validation models. The mean age of participants in the derivation models varied from 44.9 to 57.7 years, and the proportion of male subjects ranged from 46.8% to 70.5%.[27–30 32 33] The mean age of participants in the validation models varied from 41.8 to 57.1 years, and the proportion of male subjects ranged from 23.4% to 75.5%[25–28 30–32] (table 2; online supplemental table S3).

The number of CKD cases varied greatly in the derivation models, from 81[28] to 947[27]; the corresponding numbers in the validation models were 27[32] and 1359.[26] Of note, number of CKD cases could not be extracted from the validation work by Bradshaw *et al*.[27] The ratio of outcome events per number of candidate predictors in the derivation models ranged from 2.3[28] to 135.3.[27] This ratio could not be calculated for the derivation models by Wen *et al*[29] and Wu *et al*.[30] Across all reports, missing data were handled by conducting a complete-case analysis[25–32]; this information was not available in the study by

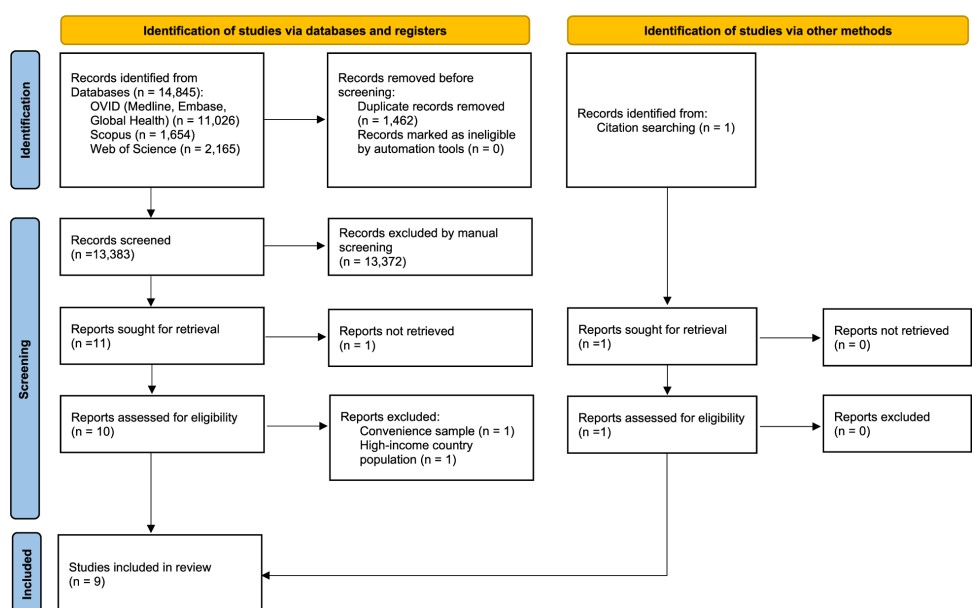

**Figure 1** PRISMA 2020 flow diagram. PRISMA, Preferred Reporting Items for Systematic Reviews and Meta-Analyses.

**Table 2** General characteristics

| No of report | Study | Country | Outcome prevalence (%) | Mean age (years) | Men (%) | Outcome details | Baseline sample size | No of outcome events | Outcome events per candidate predictors |
|---|---|---|---|---|---|---|---|---|---|
| 1 | Asgari et al, 2020[26] | Iran | 6 years validation: 22.08 9 years validation: 41.94 | 6 years validation: 46.02 9 years validation: NI | 6 years validation: 40.1 9 years validation: 40.6 | CKD was defined as eGFR <60mL/min/1.73 m², provided by the MDRD formula | 6 years validation: 3270 9 years validation: 3240 | 6 years validation: 722 9 years validation: 1359 | For every model validation: n/a |
| 2 | Bradshaw et al, 2019[27] | India | For every model validation: 10.89 For every model validation: NI | For every model derivation: 44.9 For every model validation: NI | For every model derivation: 46.8 For every model validation: NI | CKD was defined as an eGFR rate <60mL/min/1.73 m² (estimated with the CKD-EPI equation) or UACR ≥30mg/g | For every model derivation: 8698 Urban model validation: 4065 Rural model validation: 4940 | For every model derivation: 947 For every model validation: NI | Model 1 derivation: 31.6 Model 2 derivation: 41.2 Model 3a derivation: 135.3 Model 3b derivation: 118.4 For every model validation: n/a |
| 3 | Carrillo-Larco et al, 2017[28] | Peru | For every model derivation: 3.42 For every model validation: 5.41 | For every model derivation: 57.7 For every model validation: 57.1 | For every model derivation: 49.4 For every model validation: 47.7 | CKD was defined as eGFR <60mL/min/1.73 m², provided by the MDRD formula | For every model derivation: 2368 For every model validation: 1459 | For every model derivation: 81 For every model validation: 79 | Complete model derivation: 2.25 Lab-free model derivation: 3.1 For every model validation: n/a |
| 4 | Mogueo et al, 2015[25] | South Africa | For every eGFR model validation: 28.71 For every eGFR or proteinuria model validation: 29.71 | For every model validation: 55 | For every model validation: 23.4 | CKD was defined as eGFR <60mL/min/1.73m2, provided by the 4-variable MDRD formula | For every model validation: 902 | For every eGFR model validation: 259 For every eGFR or proteinuria model validation: 268 | For every model validation: n/a |
| 5 | Saranburut et al, 2017 - Framingham Heart Study[31] | Thailand | MDRD model validation: 10.37 CKD-EPI model validation: 10.01 | MDRD model validation: 54.6 CKD-EPI model validation: 54.7 | MDRD model validation: 70.8 CKD-EPI model validation: 71.5 | MDRD model validation: CKD was defined as eGFR <60mL/min/1.73 m², provided by the MDRD formula CKD-EPI model validation: CKD was defined as eGFR <60mL/min/1.73 m², provided by the CKD-EPI equation | MDRD model validation: 2141 CKD-EPI model validation: 2328 | MDRD model validation: 222 CKD-EPI model validation: 233 | For every model validation: n/a |

Continued

**Table 2** Continued

| No of report | Study | Country | Outcome prevalence (%) | Mean age (years) | Men (%) | Outcome details | Baseline sample size | No of outcome events | Outcome events per candidate predictors |
|---|---|---|---|---|---|---|---|---|---|
| 6 | Saranburut et al, 2017[31] | Thailand | For every model derivation: 8.51 For every model validation: 1.94 | For every model derivation: 51.3 For every model validation: 45.6 | For every model derivation: 70.5 For every model validation: 70.5 | CKD was defined as a preserved GFR (eGFR ≥60mL/min/1.73 m²) at baseline and subsequently developed decreased GFR (eGFR <60mL/min/1.73 m²) at the 10year follow-up, provided by the Two-level Race Variable CKD-EPI equation (using the non-black coefficient) | For every model derivation: 3186 For every model validation: 1395 | For every model derivation: 271 For every model validation: 27 | Model 1 derivation: 18.1 Model 1 BMI derivation: 18.1 Model 2 derivation: 16.9 Model 3 derivation: 12.3 For every validation model: n/a |
| 7 | Thakkinstian et al, 2011[33] | Thailand | 18.10 | 45.2 | 45.5 | CKD was defined as a combination of stages I to V. CKD stage I and II was defined as eGFR ≥90 and eGFR 60–89mL/min/1.73 m², respectively; with haematuria or UACR ≥30mg/g. CKD stage III, IV, and V was defined as eGFR 30–59, 15–29, and <15mL/min/1.73 m², respectively; regardless of kidney damage (eGFR was calculated using the MDRD formula) | 3459 | 626 | 16.9 |
| 8 | Wen et al, 2020[29] | China | For every derivation model: 18.06 | For every derivation model: 50 | For every derivation model: 44.7 | CKD was defined as an eGFR rate <60mL/min/1.73 m² (assessed with the modified Chinese MDRD equation) or UACR ≥30mg/g | For every derivation model: 3266 | For every derivation model: 590 | For every derivation model: NI |
| 9 | Wu et al, 2016[30] | China | Model derivation: 2.05 Model validation: 1.10 | Model derivation: 45.3 Model validation: 41.8 | Model derivation: 56.7 Model validation: 63.7 | CKD was defined as eGFR <60mL/min/1.73 m², eGFR <60mL/min/1.73 m² provided by the CKD-EPI equation | Model derivation: 14374 Model validation: 4371 | Model derivation: 294 Model validation: 48 | Model derivation: NI Model validation: n/a |

BMI, body mass index; CKD, chronic kidney disease; CKD-EPI, Chronic Kidney Disease-Epidemiology Collaboration; eGFR, estimated glomerular filtration rate; MDRD, modification of diet renal disease; n/a, not applicable; NI, no information; UACR, urinary albumin-to-creatinine ratio.

Thakkinstian *et al*[33] (table 2; online supplemental table S3).

## What has been done?

In 2011, Thakkinstian *et al* derived one model using cross-sectional data.[33] In 2015, Mogueo *et al* used cross-sectional data to validate two models that were previously developed in South Korea and Thailand using two different outcome definitions for each model, that is, they provided estimates for four model validations.[25] In 2016, Wu *et al* used cross-sectional data to derive and validate one model, that is, they provided estimates for two models (one derivation and one validation).[30] In 2017, Carrillo-Larco *et al* used cross-sectional data to derive and validate two models, that is, they provided estimates for four models (two derivations and two validations).[28] Saranburut *et al* prospectively validated the Framingham Heart Study risk score on a cohort using two different outcome definitions, that is, they provided estimates for two model validations.[31] Saranburut *et al* prospectively developed four models and validated two of them using cohort data, that is, they provided estimates for six models (four derivations and two validations).[32] In 2019, Bradshaw *et al* used cross-sectional data to derive four models, one of them was validated on two populations (rural and urban), that is, they provided estimates for six models (four derivations and two validations).[27] In 2020, Asgari *et al* prospectively validated a model from the Netherlands for 6- and 9 years CKD prediction, that is, they provided estimates for two model validations.[26] Wen *et al* prospectively derived two models.[29] Overall, 14 models were derived and fifteen underwent validation (hence the 29 rows in table 4).

## Outcome ascertainment

Across all reports, CKD was defined as eGFR <60 mL/min/1.73m²[25–33] assessed by either the Modification of Diet Renal Disease (MDRD) formula[25 26 28 29 31 33] or the CKD Epidemiology Collaboration (CKD-EPI) formula.[27 30–32] In addition to the eGFR assessment, Bradshaw *et al*[27] and Wen *et al*[29] defined CKD as a urinary albumin-to-creatinine ratio (UACR) ≥30 mg/g. Mogueo *et al* validations also considered CKD as any nephropathy including stages I–V of the 'Kidney Disease: Improving Global Outcomes' classification.[25] Thakkinstian *et al* also considered CKD as eGFR ≥60 mL/min/1.73 m² if it had haematuria or UACR ≥30 mg/g[33] (table 2).

## Predictors and modelling

Logistic regression analysis was conducted in all derivation models.[27–30 32 33] Selection of the final predictors was based on modelling techniques: backward[27 28] and forward selection[29 30 32 33] (online supplemental table S3). All studies categorised numerical variables. The most frequent predictors included in the models were: age, diabetes mellitus and sex (online supplemental figure S2).

## Model performance

All studies reported calibration and discrimination metrics, except for the validations by Bradshaw *et al*[27] and Carrillo-Larco *et al*[28] (online supplemental table S3). Regarding discrimination metrics, the area under the receiver operating characteristic curve and C-statistic were over 63%[31] and 70%,[27] respectively. Among all studies, sensitivity ranged from 56.8%[29] to 84.0%,[25] specificity ranged from 65.1%[29] to 86.3%,[30] positive predictive value (PPV) ranged from 8.8%[28] to 33.8%,[29] and negative predictive value (NPV) ranged from 89.4%[29] to 99.1%.[28] The NPV was the best metric, consistently above 89.4% (table 3).

## Risk of bias

All studies showed a high risk of bias due to insufficient or inadequate analytical reporting. The flaw regarding the analysis criteria can be explained by how original reports handled missing data and predictors categorisation. The participants and predictors criteria had low risk of bias in most of the reports. Most of the individual reports demonstrated an inappropriate evaluation of performance metrics.[26 28–33] Low applicability concern was noted (table 4; online supplemental table S4).

## DISCUSSION
### Main findings

This systematic review summarised all available risk scores for CKD in LMIC. In so doing, we provided the most comprehensive list of CKD risk scores to enhance primary prevention and early diagnosis of CKD in LMIC. Although the available models had acceptable discrimination metrics and, when available, acceptable calibration metrics, these models had serious methodological limitations such as a reduced number of outcome events. The best performance metric across risk scores was the NPV. Overall, CKD risk prediction tools in LMIC need rigorous development and validation so that they can be incorporated into clinical practice and interventions. The available evidence would not support using any of the available CKD risk scores across LMIC.

### Limitations of the review

We did not search grey literature. We argue that this limitation would not substantially change our results because these sources are most likely not to have included a random sample of the general population and are likely to have included a small sample size with few outcome events. That is, we would not expect to find a report in the grey literature with a much better methodology than that of the studies herein summarised.

### Limitations of the selected reports

Several LMIC do not have a CKD risk score, particularly countries in Central America and Oceania. This should encourage public health officers and researchers to develop CKD prediction models. They could conduct new

**Table 3** Performance metrics

| No | Study | Discrimination (%) | Classification measures |
|---|---|---|---|
| 1 | Asgari et al, 2020[26] | 6 years validation: AUC (95% CI) for final intercept adjusted model=Male: 76 (72 to 79) and Female: 71 (69 to 73) <br> 9 years validation: AUC (95% CI) for final intercept adjusted model=Male: 71 (67 to 74) and Female: 70 (68 to 73) | 6 years validation: For men at a cut-off of 25: sensitivity=72.7%; specificity=67.6%. For women at a cut-off of 19: sensitivity=66.8%; specificity=65.6% <br> 9 years validation: For men at a cut-off of 25: sensitivity=64.5%; specificity=69.5%. For women at a cut-off of 23: sensitivity=56.9%; specificity=76.6% |
| 2 | Bradshaw et al, 2019[27] | Model 1 derivation: C-statistic (95% CI)=79 (78 to 81) <br> Model 2 derivation: C-statistic (95% CI)=73 (72 to 75) <br> Model 3a derivation: C-statistic (95% CI)=77 (75 to 79) <br> Model 3b derivation: C-statistic (95% CI)=77 (76 to 79) <br> Urban validation: C-statistic (95% CI)=74 (73 to 74) <br> Rural validation: C-statistic (95% CI)=70 (69 to 71) | Model 1 derivation: At a cut-off of 0.09: sensitivity=72%; specificity=72%; positive predictive value=24%; negative predictive value=96% <br> Model 2 derivation: At a cut-off of 0.09: sensitivity=68%; specificity=67%; positive predictive value=20%; negative predictive value=95% <br> Model 3a derivation: At a cut-off of 0.09: sensitivity=71%; specificity=70%; positive predictive value=22%; negative predictive value=95% <br> Model 3b derivation: At a cut-off of 0.09: sensitivity=71%; specificity=70%; positive predictive value=22%; negative predictive value=95% <br> Urban model validation: NI <br> Rural model validation: NI |
| 3 | Carrillo-Larco et al, 2017[28] | Complete model derivation: AUC=76.2 <br> Lab-free model derivation: AUC=76 <br> Complete model validation: AUC=70 <br> Lab-free model validation: AUC=70 | Complete model derivation: At a cut-off of 2: sensitivity=82.5%; specificity=70.0%; positive predictive value=8.8%; negative predictive value=99.1%; likelihood ratio positive=2.8; likelihood ratio negative=0.3 <br> Lab-free model derivation: At a cut-off of 2: sensitivity=80%; specificity=72%; positive predictive value=9.1%; negative predictive value=99%; likelihood ratio positive=2.9; likelihood ratio negative=0.3 <br> Complete model validation: At a cut-off of 2: sensitivity=70.5%; specificity=69.1%; positive predictive value=11.4%; negative predictive value=97.6%; likelihood ratio positive=2.3; likelihood ratio negative=0.4 <br> Lab-free model validation: At a cut-off of 2: sensitivity=70.5%; specificity=69.7%; positive predictive value=11.6%; negative predictive value=97.7%; likelihood ratio positive=2.3; likelihood ratio negative=0.4 |
| 4 | Mogueo et al, 2015[25] | South Korean eGFR model validation: C-statistic (95% CI)=79.7 (76.5 to 82.9) <br> Thai eGFR model validation: C-statistic (95% CI)=76 (72.6 to 79.3) <br> South Korean eGFR or proteinuria model validation: C-statistic (95% CI)=81.1 (78.0 to 84.2) <br> Thai eGFR or proteinuria model validation: C-statistic (95% CI)=77.2 (73.9 to 80.5) | South Korean eGFR model validation: At a cut-off of 0.30: sensitivity=82%; specificity=67% <br> Thai eGFR model validation: At a cut-off of 0.31: sensitivity=73%; specificity=72% <br> South Korean eGFR or proteinuria model validation: At a cut-off of 0.31: sensitivity=84%; specificity=68% <br> Thai eGFR or proteinuria model validation: At a cut-off of 0.32: sensitivity=74%; specificity=73% |
| 5 | Saranburut et al, 2017 - Framingham Heart Study[31] | MDRD model validation: AUC (95% CI)=69 (66 to 73) <br> CKD-EPI model validation: AUC (95% CI)=63 (57 to 65) | MDRD model validation: NI <br> CKD-EPI model validation: NI |
| 6 | Saranburut et al, 2017 - Model 1 (derivation Clinical only)[31] | Model 1 derivation: AUC (95% CI)=72 (69 to 75) <br> Model 1 BMI derivation: AUC (95% CI)=72 (69 to 75) <br> Model 2 derivation: AUC (95% CI)=79 (76 to 82) <br> Model 3 derivation: AUC (95% CI)=80 (77 to 82) <br> Model 1 validation: AUC (95% CI)=66 (55 to 78) <br> Model 2 validation: AUC (95% CI)=88 (80 to 95) | Model 1 derivation: NI <br> Model 1 BMI derivation: NI <br> Model 2 derivation: NI <br> Model 3 derivation: NI <br> Model 1 validation: NI <br> Model 2 validation: NI |

Continued

**Table 3** Continued

| No | Study | Discrimination (%) | Classification measures |
|---|---|---|---|
| 7 | Thakkinstian *et al*, 2011 (derivation)[33] | C-statistic of internal validation=74.1 | At a cut-off of 5: sensitivity=76%; specificity=69% |
| 8 | Wen *et al*, 2020 - Simple Risk Score (derivation)[29] | Simple model derivation: AUC (95% CI)=71.7 (68.9 to 74.4)<br>Best-fit model derivation: AUC (95% CI)=72.1 (69.3 to 74.8) | Simple model derivation: At a cut-off of 14: sensitivity=70.5%; specificity=65.1%; positive predictive value=29.8%; negative predictive value=91.3%; likelihood ratio positive=2.0; likelihood ratio negative=0.5<br>Best-fit model derivation: At a cut-off of 24: sensitivity=56.8%; specificity=76.6%; positive predictive value=33.8%; negative predictive value=89.4%; likelihood ratio positive=2.4 likelihood ratio negative=0.6 |
| 9 | Wu *et al*, 2016 (derivation)[30] | Model derivation: AUC (95% CI)=89.4 (86.1 to 92.6)<br>Model validation: AUC (95% CI)=88.0 (82.9 to 93.1) | Model derivation: At a cut-off of 36: sensitivity=82%; specificity=86.3%<br>Model validation: NI |

AUC, area under the curve; BMI, body mass index; CKD-EPI, Chronic Kidney Disease-Epidemiology Collaboration; MDRD, Modification of Diet Renal Disease; NI, no information.

epidemiological studies or leverage on available health surveys with kidney biomarkers. These models could have pragmatic and direct applications in clinical medicine, by providing a tool for early identification of CKD cases. Similarly, these models could inform public health interventions and planning, by providing a tool to quantify the size of the population likely to have or to develop CKD.

Clinical guidelines state that CKD is defined as a sustained structural or functional kidney damage for ≥3 months.[34] In the studies herein summarised, CKD was defined at one point in time. Future work could expand the definition of CKD to also incorporate the lapse during which the patient had kidney damage. In addition, different procedures were used to define CKD including eGFR, proteinuria, and UACR. Even among those studies in which CKD was defined with eGFR, they used different equations to compute the eGFR. Researchers and practitioners in LMIC could agree on the best and most pragmatic as well as cost-effective definition of CKD, so that future models could use this definition. This would improve the comparability and extrapolability of the models.

All reports in which a new CKD risk score was developed selected the predictors through univariate analyses,[27–30 32 33] which is not be the best approach to choose predictors.[35–37] Ideally, predictors should be selected based on expert knowledge, or among those with the strongest association evidence with CKD. In a similar vein, predictors selection should be guided by the target population. For example, CKD prediction models for populations in LMIC should prioritise simple biomarkers or inexpensive clinical evaluations (eg, blood pressure). In this way, the risk score is likely to be used in clinical practice in resource-limited settings. Another relevant methodological limitation was how the original reports handled missing data. To the extent possible, multiple imputation should be implemented to maximise available

data and to avoid potential bias by studying only observations with complete information.

Calibration assesses the degree of agreement between actual outcomes and model prediction, whereas discrimination is the ability of the model to differentiate people with and without the outcome. Calibration metrics need to be consistently reported and should inform the direction of the miscalibration. Most of the studies used the Hosmer-Lemeshow $\chi^2$ test as the calibration metric. Unfortunately, this test does not inform on whether the model prediction is overestimating or underestimating the observed risk; calibration plots are a useful alternative. Therefore, it was not always possible to reach strong conclusions about the performance of the available models. Prognostic models should be updated before they can be applied in a new target population. This process is known as recalibration. Because we found a handful of prognostic models in some countries, it is debatable whether these can be successfully used in other populations. Available prognostic models for CKD would need to be recalibrated and independently validated in new target populations.

## Clinical and public health relevance

The Latin American Society of Nephrology and Hypertension (Sociedad Latinoamericana de Nefrología e Hipertensión) recommends to annually screen for CKD with several markers: blood pressure, serum creatinine, proteinuria and urinalysis.[38] The South African Renal Society guidelines also recommend CKD screening annually, yet they focus on high-risk populations: people with diabetes, hypertension, or HIV.[39] This recommendation is endorsed by the Asian Forum for Chronic Kidney Disease Initiatives, extending it to individuals ≥65 years, people consuming nephrotoxic substances, and those with family history of CKD and past history of acute kidney injury.[40] Although it seems reasonable to screen people with risk

**Table 4** Risk of bias (RoB) assessment of individual diagnostic/prediction models

| Study | Objective | RoB | | | | Applicability | | | Overall | |
|---|---|---|---|---|---|---|---|---|---|---|
| | | Participants | Predictors | Outcome | Analysis | Participants | Predictors | Outcome | RoB | Applicability |
| Asgari et al, 2020 European Risk Assessment tool (6 years)[26] | Validation | + | + | ? | – | + | + | + | – | + |
| Asgari et al, 2020 European Risk Assessment tool (9 years)[26] | Validation | + | + | ? | – | + | + | + | – | + |
| Bradshaw et al, 2019—model 1[27] | Derivation | + | + | ? | – | + | + | + | – | + |
| Bradshaw et al, 2019—model 2[27] | Derivation | + | + | ? | – | + | + | + | – | + |
| Bradshaw et al, 2019—model 3 a[27] | Derivation | + | + | ? | – | + | + | + | – | + |
| Bradshaw et al, 2019—model 3b[27] | Derivation | + | + | ? | – | + | + | + | – | + |
| Bradshaw et al, 2019—model 3 a (CARRS-I urban)[27] | Validation | + | + | ? | – | + | + | + | – | + |
| Bradshaw et al, 2019—model 3 a (UDAY rural)[27] | Validation | + | + | ? | – | + | + | + | – | + |
| Carrillo-Larco et al, 2017—CRONICAS-CKD (complete)[28] | Derivation | + | + | + | – | + | + | + | – | + |
| Carrillo-Larco et al, 2017—CRONICAS-CKD (lab-free)[28] | Derivation | + | + | + | – | + | + | + | – | + |
| Carrillo-Larco et al, 2017—CRONICAS-CKD (complete)[28] | Validation | + | + | + | – | + | + | + | – | + |
| Carrillo-Larco et al, 2017—CRONICAS-CKD (lab-free)[28] | Validation | + | + | + | – | + | + | + | – | + |
| Mogueo et al, 2015—South Korean model (eGFR)[25] | Validation | + | + | ? | – | + | + | + | – | + |
| Mogueo et al, 2015—Thai model (eGFR)[25] | Validation | + | + | ? | – | + | + | + | – | + |
| Mogueo et al, 2015—South Korean model (eGFR or proteinuria)[25] | Validation | + | + | ? | – | + | + | + | – | + |
| Mogueo et al, 2015—Thai model (eGFR or proteinuria)[25] | Validation | + | + | ? | – | + | + | + | – | + |
| Saranburut et al, 2017—Framingham Heart Study (MDRD)[31] | Validation | + | + | ? | – | + | + | + | – | + |
| Saranburut et al, 2017—Framingham Heart Study (CKD-EPI)[31] | Validation | + | + | ? | – | + | + | + | – | + |
| Saranburut et al, 2017—model 1 (Clinical only)[31] | Derivation | + | + | ? | – | + | + | + | – | + |
| Saranburut et al, 2017—model 1 BMI (Clinical only)[31] | Derivation | + | + | ? | – | + | + | + | – | + |
| Saranburut et al, 2017—model 2 (Clinical +Limited laboratory tests)[31] | Derivation | + | + | ? | – | + | + | + | – | + |
| Saranburut et al, 2017—model 3 (Clinical +Full laboratory tests)[31] | Derivation | + | + | ? | – | + | + | + | – | + |
| Saranburut et al, 2017—model 1 (Clinical only)[31] | Validation | + | + | ? | – | + | + | + | – | + |
| Saranburut et al, 2017—model 2 (Clinical +Limited laboratory tests)[31] | Validation | + | + | ? | – | + | + | + | – | + |
| Thakkinstian et al, 2011[33] | Derivation | + | + | ? | – | + | + | + | – | + |
| Wen et al, 2020—Simple Risk Score[29] | Derivation | + | + | ? | – | + | + | + | – | + |

Continued

**Table 4** Continued

| Study | Objective | RoB | | | | Applicability | | | Overall | |
|---|---|---|---|---|---|---|---|---|---|---|
| | | Participants | Predictors | Outcome | Analysis | Participants | Predictors | Outcome | RoB | Applicability |
| Wen et al, 2020—Best-fit Risk Score[29] | Derivation | + | + | ? | − | + | + | + | − | + |
| Wu et al, 2016[30] | Derivation | + | + | ? | − | + | + | + | − | + |
| Wu et al, 2016[30] | Validation | + | + | ? | − | + | + | + | − | + |

Prediction model Risk Of Bias ASsessment Tool[20,21]; RoB, + indicates low RoB/low concern regarding applicability; − indicates high RoB/high concern regarding applicability; and ? indicates unclear RoB/unclear concern regarding applicability.
BMI, body mass index; CKD-EPI, Chronic Kidney Disease-Epidemiology Collaboration; eGFR, estimated glomerular filtration rate; MDRD, Modification of Diet Renal Disease.

factors such as hypertension and diabetes, this approach may miss a large proportion of the high-risk population because they could be unaware of their condition.[41 42] In this case, risk scores could be useful because they can be applied to large populations regardless of whether they are aware of their hypertension or diabetes status. Unfortunately, our work would not support nor encourage the inclusion of available risk scores for CKD in clinical guidelines in LMIC. Instead, our results urgently call to improve risk prediction research in LMIC. Therefore, CKD risk scores could be included into clinical practice to identify high-risk individuals and to inform the patient's management plan as is the case in other fields such as cardiovascular primary prevention.

## CONCLUSIONS

This systematic review of diagnostic and prognostic models of CKD did not find conclusive evidence to recommend the use of a single CKD score across LMIC. Nonetheless, we identified relevant efforts in Iran, India, Peru, South Africa, China and Thailand; these models would require further external validation before they can be applied in other LMIC. We encourage researchers and practitioners to develop and validate CKD risk scores, which are cost-efficient tools to early identify CKD prevalent and incident cases so that they can receive timely treatment.

**Author affiliations**
[1]School of Medicine 'Alberto Hurtado', Universidad Peruana Cayetano Heredia, Lima, Peru
[2]CRONICAS Centre of Excellence in Chronic Diseases, Universidad Peruana Cayetano Heredia, Lima, Peru
[3]Health Innovation Laboratory, Institute of Tropical Medicine 'Alexander von Humboldt', Universidad Peruana Cayetano Heredia, Lima, Peru
[4]Emerge, Emerging Diseases and Climate Change Research Unit, School of Public Health and Administration, Universidad Peruana Cayetano Heredia, Lima, Peru
[5]Department of Epidemiology and Biostatistics, School of Public Health, Imperial College London, London, UK

**Contributors** RMCL, DJA-G and EJA conceived the idea. RMCL, DJA-G and EJA conducted the search. DJA-G and EJA wrote the manuscript. All authors approved the submitted version and are responsible of its content. All authors act as guarantors.

**Funding** RMCL is supported by a Wellcome Trust International Training Fellowship (214185/Z/18/Z).

**Map disclaimer** The inclusion of any map (including the depiction of any boundaries therein), or of any geographic or locational reference, does not imply the expression of any opinion whatsoever on the part of BMJ concerning the legal status of any country, territory, jurisdiction or area or of its authorities. Any such expression remains solely that of the relevant source and is not endorsed by BMJ. Maps are provided without any warranty of any kind, either express or implied.

**Competing interests** None declared.

**Patient consent for publication** Not applicable.

**Ethics approval** This review was deemed as a low risk because human subjects were not directly involved.

**Provenance and peer review** Not commissioned; externally peer reviewed.

**Data availability statement** Data sharing not applicable as no datasets generated and/or analysed for this study.

**ORCID iDs**
Diego J Aparcana-Granda http://orcid.org/0000-0001-9993-0029
Edson J Ascencio http://orcid.org/0000-0002-8340-9236
Rodrigo M Carrillo Larco http://orcid.org/0000-0002-2090-1856

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
