## [Reviewer comments · BMJ Open]

ARTICLE DETAILS

TITLE (PROVISIONAL)	A systematic review of diagnostic and prognostic models of chronic kidney disease in low- and middle- income countries
AUTHORS	Aparcana-Granda, Diego J.; Ascencio, Edson J.; Carrillo Larco, Rodrigo M.

VERSION 1 – REVIEW

REVIEWER	Tope Oyelade UCL, Institute for Liver and Digestive Health
REVIEW RETURNED	25-Dec-2021

GENERAL COMMENTS	A well planned, executed, and presented systematic review on the diagnostic and prognostic models for chronic kidney disease (CKD) in low- and middle-income countries (LMICs). Following series of reporting and assessment guidelines, and systematic screening, synthesis and review of literature, authors presented the used models and showed that age, diabetes mellitus and sex were the main predictors of risk of CKD according to he reported models. Authors also showed that evidence of diagnostic and prognostic models for CKD is in general weak, and inconsistent in LMICs. And recommend research in this area to develop consistent, accurate models for the diagnosis and prognosis of CKD in LMICs. Below are some recommendations: Major 1. Tables 2 and 3 are cumbersome with too many rows containing repetitive data. The extra information (validation/ derivation etc) or data could be squeezed into single cells so that each row contains single authors and data/information clearly defined in cells within that column. This will make for a better reading experience for potential readers. The advantage of systematic reviews is the ability to concisely present the summary findings.2. The manuscript is laden with typos and grammatic errors and is recommended for thorough proofing (few details below).3. Page 6/ Study selection: Authors stated exclusion of studies using models developed by machine learning techniques. However, the justification for this decision is not clear.4. Page 9, Line 42: do you mean "sensitivity"? If yes, please check tables 2 and 3 and correct as needed. Minor 1. Page 3/ Line 15-16: "risk of bias assessment"2. Page 4: It might be clearer to separate the strengths and limitation by subheading. This will help clarify which is which.3. Page 4/ Line 11: "for in the LMIC population"4. Page 5/ Line 23: "allow screening of large populations"5. Page 5/ Line 49: "Checklist"
--

	6. Page 6, Lines 10-12: "We sought models which assessed the current CKD status (i.e., diagnostic) or future CKD risk (i.e., prognostic), aiming to..." 7. Page 7, Line 26: "Results and opinions expressed in the article are entirely the authors". 8. Page 7/ Lines 38-39: "population was selected by convenience" reads better as "population was not randomly selected". 9. Page 7/ Line 58: "...most recent study was published in 2018..." 10. Page 8, From line 28: Could you present this description in chronological order? This will allow for clarity and help readers understand the directions and trends of model derivation and validation. 11. Page 11, Line 53-55: "recommends to annually screen for CKD with several studies" is not clear. Do you mean "recommends to annually screen for CKD with several markers"? 12. Page 12/ Line 67: As is the case. 13. The mining of 13,383 studies from the search might be an implication of a very vague search strategy. It will be interesting to know why out of such huge search result, only a final 11 studies were relevant to this review. And also does spending so much time and resources on the screening not have any effect on the quality of the study?
--	--

REVIEWER	Ikechi G Okpechi University of Cape Town, Division of Nephrology and Hypertension
REVIEW RETURNED	03-Jan-2022

GENERAL COMMENTS	This study by Aparcana-Granda et al sought to summarize available chronic kidney disease diagnostic and prognostic models in Low- and Middle-Income countries through a systematic review of available evidence. Unfortunately, there isn't much evidence available from these countries and of those available, the evidence to use these models have been found to be, at best, very weak and unsupported. Their paper is well written, however, I feel that the discussion around the model performance needs to be expanded as this will give better understanding of why the authors have arrived at their conclusions.
--

VERSION 1 – AUTHOR RESPONSE

Reviewer 1: Mr. Tope Oyelade, UCL

Q1. Tables 2 and 3 are cumbersome with too many rows containing repetitive data. The extra information (validation/ derivation etc) or data could be squeezed into single cells so that each row contains single authors and data/information clearly defined in cells within that column. This will make for a better reading experience for potential readers. The advantage of systematic reviews is the ability to concisely present the summary findings.

A1. We updated Table 2 and Table 3 so that they only show one study per row.

Q2. The manuscript is laden with typos and grammatic errors and is recommended for thorough proofing (few details below).

A2. We proofread the manuscript and corrected all errors, including those kindly highlighted by the reviewer.

Q3. Page 6/ Study selection: Authors stated exclusion of studies using models developed by machine learning techniques. However, the justification for this decision is not clear.

A3. We excluded models developed with machine learning techniques because they tend to report incomplete information about the performance metrics; this has been previously pointed out.^{1 2} In

addition, guidelines for reporting systematic reviews of prognostic models are been updated to also include machine learning models, though these are not available yet.³ We elaborated on this matter on page 05: *"We also excluded models that were developed using machine learning techniques due to their usually poor report of performance metrics, as noted from previous reviews. To overcome this limitation, CHARMS and PROBAST tools are currently being adapted to machine learning methodology but are yet to be published."*

Q4. Page 9, Line 42: do you mean "sensitivity"? If yes, please check tables 2 and 3 and correct as needed.

A4. We changed the word "sensibility" for "sensitivity" throughout the manuscript where needed.

Q5. Page 3/ Line 15-16: "risk of bias assessment".

A5. We have changed the word as suggested.

Q6. Page 4: It might be clearer to separate the strengths and limitations by subheading. This will help clarify which is which.

A6. We accepted the recommendation and added the subheadings to better differentiate "strengths" from "limitations".

Q7. Page 4/ Line 11: "for in the LMIC population"

A7. We corrected this sentence.

Q8. Page 5/ Line 23: "allow screening of large populations"

A8. We changed the sentence as suggested.

Q9. Page 5/ Line 49: "Checklist"

A9. If the reviewer kindly allows, we will rather keep "Checklist" as is. "Checklist" contains two capitalized letters as in the original acronym of the CHARMS tool.

Q10. Page 6, Lines 10-12: "We sought models which assessed the current CKD status (i.e., diagnostic) or future CKD risk (i.e., prognostic), aiming to..."

A10. We changed this sentence as suggested.

Q11. Page 7, Line 26: "Results and opinions expressed in the article are entirely the authors".

A11. We changed the sentence as suggested.

Q12. Page 7/ Lines 38-39: "population was selected by convenience" reads better as "population was not randomly selected". Page 7/ Line 58: "...most recent study was published in 2018..."

A12. Both sentences were changed as suggested.

Q13. Page 8, From line 28: Could you present this description in chronological order? This will allow for clarity and help readers understand the directions and trends of model derivation and validation.

A13. Following the reviewer's suggestion, we edited this paragraph, and the results are now presented chronologically.

Q14. Page 11, Line 53-55: "recommends to annually screen for CKD with several studies" is not clear. Do you mean "recommends to annually screen for CKD with several markers"?

A14. We edited this sentence as suggested.

Q15. Page 12/ Line 67: As is the case.

A15. We updated the sentence as suggested.

Q16. The mining of 13,383 studies from the search might be an implication of a very vague search strategy. It will be interesting to know why out of such huge search result, only a final 11 studies were relevant to this review. And also does spending so much time and resources on the screening not have any effect on the quality of the study?

A16. To cover as many records as possible, and to decrease the probability of missing relevant publications, we conducted a comprehensive search. Our search strategy is like those used in other reviews on prognostic models, signaling a common set of terms frequently used in these reviews.

We screened 13,383 scientific papers and included 11 in the review; that is, the ration search-included was 1,216. Notably, this ratio was similar, and even smaller, than the ratio in other systematic reviews which also followed the CHARMS and PROBAST guidelines (please, refer to the table below for some examples).⁴⁻⁸

Table 1. Number of studies in the search strategy of other systematic reviews.

Study	Number of articles in the first search	Number of articles included in the review	Ratio between first search and final included studies
Zhang et al , 2020	29,668	11	2,697
Daines et al , 2019	13,180	8	1,647
Van Acker et al , 2021	13,140	8	1,642
Kim et al , 2021	30,450	21	1,450
Li et al , 2021	8,776	13	675

Overall, we do not consider the large number of initial results a limitation of our work. We believe our thorough search, which was informed and leveraged on previous research in the field is a strength of our work. Moreover, the large number of initial results should not have affected the quality of our work, considering we conducted this research in, approximately, 7 months. Enough time to carefully screen the 13,383 search results.

Reviewer 2: Dr. Ikechi G Okpechi, University of Cape Town

Q1. Their paper is well written; however, I feel that the discussion around the model performance needs to be expanded as this will give better understanding of why the authors have arrived at their conclusions.

A1. We discussed calibration and discrimination, key concept in prognosis research. These concepts are relevant to support our conclusion that it is not possible to strongly recommend any of the available prognostic models for CKD. This, because available models did not report much information on calibration metrics. We also discussed about recalibration. That is, the process by which prognostic models are updated for a new target population. This supports our conclusion that the handful of models found in some countries need recalibration and independent external validation before they can be used in new target populations (i.e., other countries). The new paragraph on page 10 reads: *“Calibration assesses the degree of agreement between actual outcomes and model prediction, whereas discrimination is the ability of the model to differentiate people with and without the outcome. Calibration metrics need to be consistently reported and should inform the direction of the miscalibration. Most of the studies used the Hosmer-Lemeshow X2 test as the calibration metric. Unfortunately, this test does not inform on whether the model prediction is overestimating or underestimating the observed risk; calibration plots are a useful alternative. Therefore, it was not always possible to reach strong conclusions about the performance of the available models. Prognostic models should be updated before they can be applied in a new target population. This process is known as recalibration. Because we found a handful of prognostic models in some countries, it is debatable whether these can be successfully used in other populations. Available prognostic models for CKD would need to be recalibrated and independently validated in new target populations.”*

VERSION 2 – REVIEW

REVIEWER	Tope Oyelade UCL, Institute for Liver and Digestive Health
REVIEW RETURNED	23-Feb-2022
GENERAL COMMENTS	Congratulation on this thorough systematic review of the models for prognostication of chronic kidney disease in low and middle income countries. Hopefully, this will encourage discussion towards development of comprehensive prognostic models to prevent the burden of CKD especially in LMICs.